# Mental Health in New Mothers: A Randomised Controlled Study into the Effects of Dietary Flavonoids on Mood and Perceived Quality of Life

**DOI:** 10.3390/nu13072383

**Published:** 2021-07-13

**Authors:** Katie Louise Barfoot, Rachel Forster, Daniel Joseph Lamport

**Affiliations:** School of Psychology & Clinical Language Sciences, University of Reading, Earley Gate, Reading RG6 7BE, UK; r.forster@student.reading.ac.uk (R.F.); daniel.lamport@reading.ac.uk (D.J.L.)

**Keywords:** mothers, mood, anxiety, mental health, women’s health, postnatal depression, postpartum, polyphenols, diet

## Abstract

The postnatal period is a significant period of physical, physiological and psychological change for mothers, rendering them particularly vulnerable to changes in mood or disorders such as postnatal depression (PND). Previous interventions with foods high in flavonoids have demonstrated beneficial acute and chronic mood effects in healthy child, adolescent and adult populations. It is unclear whether mood effects persist in populations who are potentially at-risk of developing mood disorders, such as postnatal mothers. This exploratory study investigated the effects of a 2-week daily dietary flavonoid intervention on mood (PANAS-NOW), anxiety (STAI), depressive symptoms (PHQ-8) and perceived quality of life (WHOQOL-BREF) in forty-one new mothers in the 0–12-month postnatal period, before and after flavonoid intervention. Mothers either added high flavonoid foods to their daily diet, or did not include additions following a randomised, between-groups, controlled design. Significant effects were observed in the flavonoid group with mothers reporting lower state anxiety and higher perceived quality of physical health at the 2-week timepoint. These findings suggest that regular dietary consumption of flavonoids may benefit mothers’ anxiety and perceived quality of life in the postnatal period. Replication of these results may indicate the potential for dietary flavonoids to promote healthy mood regulation in mothers or prevent the onset or severity of symptoms in postnatal psychological disorders, both of which would be beneficial for women’s health services and public mental health.

## 1. Introduction

The beneficial effects of polyphenols on health ([1,2,3] for reviews) and cognition ([4,5,6,7,8] for reviews) have been well reported within the literature. Recent investigations have focused on the impact of polyphenols on mood and have discovered advantageous effects in healthy child [9], adolescent [10] and adult [9,11] populations after polyphenol-rich food interventions. Such findings suggest that consuming foods high in flavonoids may improve or maintain positive mood in healthy populations. This also holds promise for populations with symptoms of low mood, or for those ‘at risk’ of developing mental health conditions such as postnatal depression (PND) or anxiety, both of which are rising public health concerns. Dietary intervention, such as flavonoids, may offer natural protection from symptom onset or lessen symptom severity. The current exploratory study aimed to assess mood, depression, anxiety and perceived quality of life before and after a 2-week high flavonoid dietary intervention in a sample of new mothers who had an infant under 1 year old.

PND is defined as an episode of major depressive disorder (MDD) commonly experienced by parents following the birth of their baby and affects approximately 1 in 10 new parents [12]. PND can affect both parents; around 8–10% of new fathers have been reported to develop PND in the first year following their baby’s birth [13] with yearly prevalence in mothers estimated at 10–15% [14]. Concerningly, it has been indicated that prevalence in mothers may be as high as 60% worldwide [15]. These statistics indicate PND as a serious clinical issue for postnatal healthcare services [16] and investigation into early interventions to prevent PND onset or lessen severity is warranted.

PND is distinct from postpartum blues (PPB; ‘baby blues’); the latter is a state experienced by 50–80% of mothers in the first few days following childbirth which includes symptoms such as crying, decreased appetite, mood swings, worrying and exhaustion. PPB is generally considered a ‘normal’ experience for new mothers by healthcare professionals; indeed, symptoms usually resolve within 2 weeks of birth and do not appear to negatively impact long-term maternal function [17]. PND, however, is classified as an episode of MDD that lasts longer than 2 weeks, generally occurs within 6 months of birth and negatively affects daily functioning [18,19]. Symptoms of PND are similar to MDD and include low mood, fatigue, feelings of guilt, inability to concentrate, mood swings, difficulty bonding with one’s baby, withdrawal from family and friends, irrational anger, intense irritability and thoughts of harming oneself or one’s baby [19], and research has shown that mothers with PND experience more cognitive, behavioural and interpersonal difficulties, including lower mood, energy and concentration than mothers without PND [20]. Such symptoms can significantly affect mothers’ perceived quality of life and ability to cope with daily living; in severe cases it can lead to psychosis or suicide, the latter being the most common form of maternal death in the postnatal period [21]. It is therefore imperative that persistent low mood in populations at risk of developing a depressive disorder is addressed, and a range of preventative treatments are investigated to reduce mental health disturbances in new mothers.

PND has been the focus of postnatal mental health research, most likely due to its prevalence and similarities with MDD in terms of diagnosis and treatment. It is, however, critical to assess other postnatal affective states or psychological disorders, such as anxiety, so that these can be distinguished and treated separately [22]. The current study will assess current mood, anxiety and quality of life outcomes in addition to a measure of depression so that the impact of dietary intervention can be assessed across a wider range of mood states and lifestyle factors. This is important so that other mood changes can be captured independently of PND, which may aid treatment specificity for other disorders. Current treatments for mood disturbances in the postnatal period (i.e., PND, anxiety) include psychological therapy and antidepressants, however both have varying success rates [23]. There is a pressing need to investigate natural alternatives to medication which are cost-effective, easy to access and easy to self-implement.

Flavonoids are naturally occurring plant compounds, commonly found in berries, citrus fruits, vegetables, cocoa, red wine, tea and coffee. Research has found that long-term consumption of flavonoids may support mental health and well-being. For instance, epidemiological research has found that consistent consumption of food and drink items high in flavonoids, such as fruits, vegetables and tea are predictive of reduced depression risk in later life [24]. In a sample of adults living in southern Italy, Godos et al. [25] found that higher dietary intake of foods abundant in specific flavonoid subclasses (phenolic acid, anthocyanins and flavanones) was significantly related to lower depressive symptoms. Symptoms reduced in a dose-response manner to higher levels of anthocyanins and flavanones, suggesting that high levels of particular flavonoid subclasses may underlie observed mood effects. A recent review [26] identified 29 studies where symptoms of depression in young adult populations were alleviated following polyphenol intervention, indicating a clear link between high consumption levels and improved mood outcomes. It is interesting to note that there are a number of studies showing that dietary sources of anthocyanins or flavanones have beneficial effects on chronic mood [10,11]. For example, Pase et al. [11] reported significantly enhanced feelings of calmness and contentedness in middle-aged adults (40–65 years) following a 30-day daily cocoa intervention. Across a similar timeframe, positive mood effects were also observed in an adolescent population (12–17 years), with participants reporting significantly lower depressive symptoms at the end of a 4-week daily wild blueberry intervention [10]. Improvements in quality of life have also been observed in a depressed sample (50–55 years) following a 12-week daily soy and resveratrol intervention [27]. Such results suggest that daily supplementation with flavonoid-rich foods or supplements may have a positive impact on mood, quality of life or depressive symptoms in healthy and depressed individuals.

Mechanistically, these findings are supported by animal research that suggests that flavonoids may alleviate symptoms of anxiety via neuronal signaling pathways ([28] for review). Allam et al. [29] conducted a study in rats where oxidative stress (OS) was chemically induced, and high anxiety-like behaviours were subsequently observed. Following a 3-week daily grape powder intervention (15 g/L), OS grape-fed rats showed significantly fewer anxiety-like behaviours compared to OS control-fed rats, at a level similar to healthy (non-OS-induced) control and grape-fed rats. This implies that an absence of OS may be associated with low levels of anxiety, or that OS results in higher levels of anxiety. Positive changes in extracellular signal-regulated kinase 1/2 (ERK-1/2), cAMP-response element-binding protein (CREB) and brain-derived neurotrophic factor (BDNF) were also observed in OS grape-fed rats, to a similar degree as control-fed and grape-fed rats. Such activation of signaling pathways was not observed in OS control-fed rats. These findings imply that flavonoids may be able to reduce anxiety symptom intensity by reducing OS, potentially through neuronal signaling mechanisms. Similar ameliorations in chronic stress-induced behaviours have been observed in rodents post-kaempferol intervention (the major flavonoid compound in broccoli, tea and strawberries; [30]), believed to be modulated by reductions in OS, inhibition of inflammatory cytokines and up-regulation of AKT/B-catenin cascade. Hesperidin, a flavonoid found in citrus fruits, has been reported to reduce depressive-like behaviours in mice through kappa-opiod (involved in dopamine transmission; [31]) or serotonergic pathways [32]. Such findings hold implications for the potential for flavonoids to exert effects on mood, specifically anxiety, using similar neuronal mechanisms as cognitive change [33,34,35,36]. There are a number of other mechanisms by which flavonoids may exert mood effects such as changes in cerebral blood flow, transmission of GABA or monoamine oxidase inhibition. 

In light of the positive indications for mood and well-being after consumption of flavonoid-rich foods (particularly in domains associated with low mood or depression), exploring this relationship in a human sample at risk of mood disorders, such as new mothers, merits investigation. Therefore, this exploratory study aimed to investigate whether the addition of high flavonoid foods in the diet across a 2-week period can impact maternal mental health, specifically mood and perceived quality of life in the 0–12 month postnatal period, relative to a control group. It was hypothesised that mothers in the flavonoid diet group may report reduced symptoms of depression and anxiety, and increased positive mood and perceived quality of life after the 2-week intervention.

## 2. Materials and Methods

This research project was reviewed and given a favourable ethical opinion for conduct by the University of Reading School of Psychology and Clinical Language Sciences Ethics Committee (2020-075-KB).

### 2.1. Sample

Forty-one mothers were recruited via mother and baby pages on social media within a 6-week period, of which 38 mothers completed demographic information. The sample was 84% white; 11% Asian or Asian British; 3% Black/African/Caribbean/Black British; 3% Latino or Hispanic, educated (49% further; 38% higher) with most women in a domestic partnership (82%). Annual household income varied across the sample; 32% of households reported above average income (>£41,000), 37% fell in line with the current UK average (£31–40,000; UK average = £37,108 [37]) and 26% reported below average income (<£30,000). This was unsurprising as it was expected that a household with a new baby may have reduced working hours; indeed 58% of women were on maternity leave with only 34% in employment at the time of the study (full time = 8%; part time = 13%; self-employed = 13%). No women reported unemployment and 8% of women were students.

All women had a baby under the age of 12 months, with baby’s age ranging from 5 to 48 weeks. For 28 women this was their first child, whereas 9 women indicated that they also had other children (number of other children mode = 1, range 1–3). The majority of women (82%) stated that they had childcare support (e.g., nearby family) with only 18% reporting no support outside of the family home.

Eleven women had a psychological diagnosis: 5 with anxiety, 3 with PND, 2 with anxiety and depression and 1 with anxiety and eating disorder. Of these 11 women, 7 were on antidepressant selective serotonin reuptake inhibitor (SSRI) medication. The remaining 4 women with diagnoses were not on medication. It was important to include individuals with psychological diagnoses (28.95%) in the study as this was thought to reflect similar proportions in the postnatal population at the time (29.8% anxiety and/or depression diagnoses, Spring/Summer 2020 [38]). Nine women reported physical health diagnoses: 5 with asthma, 1 with diabetes, 1 with Crohns disease, 1 with Hodgkin lymphoma and 1 with asthma and arthritis. Similarly, individuals with physical diagnoses were included to reflect real life prevalence.

An a-priori power calculation using GPower 3.1 [39] revealed that 44 participants were required to achieve a power of 0.8 using an effect size (Cohen’s *d*) of 0.22 (F(1,42) = 4.07). This effect size was used based on prior research that has shown significant flavonoid-related mood effects [9,10]. A post-hoc power analysis confirmed that a sufficient power of 0.78 was achieved with the current sample size of 41 (F(1,39) = 4.09). There were no significant differences in demographic variables between the flavonoid and control group (see Table 1).

### 2.2. Design

Participants were randomly assigned (using a random number generator) to either a ‘flavonoid’ group (*n* = 21) or a ‘control’ group (*n* = 20) by the researcher. The flavonoid group were instructed to add at least one high flavonoid food item from a specified list to their daily diet for 2 weeks. The control group were not given instructions and so made no changes to their current diet. Mood, anxiety, depression and quality of life measures were recorded at baseline and at the end of the 2-week intervention in both groups.

### 2.3. Intervention

The flavonoid group were encouraged to continue eating as normal and to add at least 1 high flavonoid food item per day (from a designated list) to their current diet. The list of high flavonoid food items included ‘berry fruits (~120 g) e.g., blueberries, raspberries, strawberries, blackberries, blackcurrants, mixed berries’, ‘2 large squares of (min. 70% cocoa) dark chocolate’, ‘4–5 cups of black/green tea or caffeinated/decaffeinated coffee’, ‘1 large glass of red wine (250 mL)’, ‘1 portion of leafy green vegetables such as spinach or cabbage (~70 g)’ and ‘1 glass (250 mL) of fresh orange or grapefruit juice (not from concentrate)’. These items were chosen from the USDA Database for the Flavonoid Content of Selected Foods [41] for their high flavonoid content, likeability, accessibility and affordability in a sample that were likely to be time- and money-limited. Portion guidelines were decided based on average adult consumption of these foods in one sitting (or per day for tea and coffee). These items were also selected to cover a breadth of flavonoid subclasses in foods that are realistically consumed in the average diet. It was important to use food items and portion sizes that were agreeable, achievable and were indicative of real-life consumption.

Participants in the flavonoid group completed a food log every day of the 2-week intervention to record the flavonoid food item(s) added to their diet. According to food logs, participants consumed an average of 15.45 (SD = 4.33) high flavonoid items over the 14-day intervention period and missed, on average, 2 out of 14 days (SD = 2.10), which suggests good overall compliance (86%). Within the high flavonoid diet group, coffee was the most consumed food, followed by berry fruits. A full breakdown of high flavonoid food additions in the high flavonoid group can be seen in Table 2.

Participants in the control group did not receive any instructions regarding changes to their diet and so continued consuming their typical diet. This group did not complete flavonoid food logs as they were not required to add items to their diet. These participants were aware that the study was investigating diet and mental health in new mothers and at the end of baseline were told they would be invited to complete questionnaires again in 2-weeks’ time.

Participant blinding of each group was achieved as the study was advertised as an investigation into “diet and mothers’ mental health”. The groups were told that some participants may be asked to include additional common food items to their diet over a 2-week period, meaning they were aware that there were two separate groups but did not know the exact dietary focus of the intervention. The recruiting researcher was not blind to participant group membership due to them needing to know which information to send throughout the course of the study (e.g., high flavonoid diet sheets, food logs). This researcher only ever engaged with participants via email and the wording in these emails was standardised for consistency. Data analysis was completed fully blinded, by a different researcher.

### 2.4. Measures

All measures were administered online using Microsoft Forms. Due to the number of questionnaire items across the study, measures were distributed across two online links and were labelled ‘Part 1’ and ‘Part 2’. Participants were instructed to complete these parts consecutively.

#### 2.4.1. Habitual Flavonoid Consumption

A 35-item Food Frequency Questionnaire (FFQ) was administered at baseline to record individuals’ habitual flavonoid and junk food intake (see Table 1). Habitual flavonoid intake was derived from 22 items taken from the European Prospective Investigation of Cancer Food Frequency Questionnaire (EPIC-FFQ [42]): leafy green vegetable soups, tea (black, green, fruit, white, other), coffee, cocoa, red wine, beer/lager/cider, pure fruit juice (100%), apples, oranges/satsumas/mandarins, grapefruit, strawberries/raspberries/blueberries, dried fruit e.g., raisins, cranberries, blueberries, broccoli/spring greens/kale, leeks, onions and tomatoes. Participants rated how often they consumed each item on a 7-point Likert scale: (1) Never or less than once/month, (2) 1–3 times per month, (3) 1–2 times per week, (4) 3–5 times per week, (5) Once per day, (6) 2–3 times a day, (7) 4+ times a day. Each participant response was converted to average number of flavonoid food items per day for each participant. Junk food items (not analysed) were added as a ‘filler’, so participants believed they were answering a general diet questionnaire rather than one specific to flavonoid intake.

#### 2.4.2. Mood

The Positive and Negative Affect Schedule (PANAS-NOW; [43]) was used to assess participants’ current mood (full details in [9]). The PANAS has shown to be sensitive to flavonoid effects previously [9] and is a quick and reliable measure. Scores of positive affect (PA) and negative affect (NA) were calculated (range 20–80), where a higher score was indicative of higher PA or NA.

#### 2.4.3. Anxiety

Participants completed the State-Trait Anxiety Inventory (STAI; [44]) which consisted of 40 items. Twenty items of the STAI related to ‘state’ anxiety—the presence and severity of current anxiety symptoms. All items were rated on a 4-point Likert scale. On ‘state’ anxiety items, responses were ‘Not at all; Somewhat; Moderately so; Very much so’ and were indicative of how participants felt ‘at this moment’. On ‘trait’ anxiety items, responses applied to the frequency of feelings ‘in general’: ‘Almost never; Sometimes; Often; Almost always’. Items were appropriately reverse coded and summed to create ‘state’ anxiety (SA) and ‘trait’ anxiety (TA) scores (range 20–80; [45]). Higher SA and TA scores were indicative of higher state and trait anxiety, respectively.

#### 2.4.4. Depression

The Patient Health Questionnaire 8 (PHQ-8; [46]) was administered as a measure of depressive symptoms. The PHQ-8 has been found to be a reliable measure of current depression in the general population [47] and is appropriate for use in a non-clinical sample. The questionnaire includes 8 items that relate to the Diagnostic and Statistical Manual (DSM-IV) criteria for depression, excluding suicidal ideation. Participants rated how often they were bothered by each of the 8 items during the past 2 weeks on 4-point Likert scales. Responses were logged from ‘0—Not at all’ to ‘3—Nearly every day’. An overall score was derived by summing the responses from the 8 items (range 0–24). Cut-off scores were used to categorise the severity of depression symptoms: 5 = mild depression, 10 = moderate depression, 15 = moderately severe depression and 20+ = severe depression [46].

#### 2.4.5. Perceived Quality of Life

The World Health Organization Quality of Life (WHOQOL-BREF; [48]) questionnaire was also used in the current study. The WHOQOL-BREF is an abbreviated version of the WHOQOL-100, a well-validated, international questionnaire that allows detailed assessment of perceived quality of life. The WHOQOL-100 was deemed too lengthy and time consuming for use in the current study. The WHOQOL-BREF has been deemed suitable for investigating changes in quality of life across intervention studies lasting a minimum of 2 weeks making it appropriate for use in the current study.

The WHOQOL-BREF contained one item from each of the 24 facets of the WHOQOL-100 that relate to aspects of perceived quality of life including energy, fatigue, sleep, rest, bodily image, self-esteem, personal relationships, social support, financial resources, health and social care and home environment, and two additional items from overall quality of life and general health facets. Questionnaire items were rated using 5-point Likert scales where ‘1’ represents ‘Very poor; Very dissatisfied; Not at all; Never’ and ‘5’ represents ‘Very good; Very satisfied; An extreme amount; Extremely; Completely; Always’. Items were summed and standardised appropriately into four domains: Physical Health, Psychological, Social Relationships and Environment. A higher score was indicative of a higher perception of aptitude or quality in each domain.

### 2.5. Procedure

Participants completed assessments of mood and quality of life on two separate occasions, at baseline (day 0) and 2 weeks later. At baseline, participants completed a demographics questionnaire about themselves and their baby, which was labelled as optional due to the personal nature of some questions. Several risk factors for PND have previously been identified including pregnancy complications, prematurity of baby and maternal age (adolescents and women over 35 years having higher risk [49]). Other risk factors include being single [50] and having multiple children [51]. It was important that these measures were captured to characterise the sample, so mothers were asked their age, marital status, number of other children, pregnancy complications and their baby’s term at birth. Other demographic questions included gender, ethnicity, education, employment, household income, psychological and physical health diagnoses, medication, average sleep per night (hours), nearby childcare support and birth or breastfeeding complications. Participants also answered various questions about their baby: age, sex and feeding method (breast milk, formula or combination) at 6 and 12 months (see ‘2.1 Sample’ and Table 1).

Participants answered questionnaires in the following order: PANAS-NOW, FFQ, PHQ-8, WHOQOL-BREF and STAI. Completion of the baseline session indicated the start of the 2-week intervention for that participant. At the end of the baseline session, participants in the flavonoid condition were presented with instructions on consuming additional food items for 2 weeks. Instructions also contained a 14-day food log where participants were required to fill in which additional food item(s) from the list they had added to their daily diets. If participants missed a day, they were encouraged to record the day as ‘missed’ and continue with the diet as normal the next day. Participants in the flavonoid condition were emailed on day 5 of the intervention to check progress and to remind them to complete food logs.

Participants who were assigned to the control condition did not receive instructions nor a 14-day food log. Upon completing the baseline session, control participants were told they had completed the first half of the study and would be invited to take part in the second part of the study in 2 weeks’ time.

Participants in both conditions were emailed on day 10 of their respective intervention periods with the link to the second part of the study. They were encouraged to complete this when convenient on day 14. Follow-up email reminders were sent on day 14 and on day 16 for those who had not yet completed the second part by the end of day 14.

The study was conducted across a 6-week period between July and August 2020, after which funding for the project ceased, marking the end of the trial. Participants completed staggered baseline sessions across weeks 0–4 with intervention commencing immediately for 2 weeks. Participants were to cease consumption of additional food items (if in the flavonoid group) and complete follow-up questionnaires 2 weeks after their individual baseline session. Week 4 was the final week for recruitment of new participants to allow all participants to complete the 2-week trial in the 6-week period.

### 2.6. Statistical Analyses

Some missing data were evident due to participants only completing Part 1 or moving on to Part 2 without correct submission of Part 1 questionnaire responses. At baseline, 2 data points were missing from the PANAS (flavonoid (F) = 1; control (C) = 1), 7 from the STAI (F = 5; C = 2), 7 from the PHQ-8 (F = 5; C = 2) and 7 from the WHOQOL-BREF (F = 5; C = 2). At 2 weeks, 5 were missing from the PANAS (F = 3; C = 2), 4 from the STAI (F = 2; C = 2), 5 from the PHQ-8 (F = 3; C = 2) and 5 from the WHOQOL-BREF (F = 3; C = 2).

Data were analysed using SPSS Statistics version 24.

Independent samples t-tests and chi-square (where appropriate) were performed to assess whether there were significant group differences for demographic and baseline data.

Outcome data from the PANAS-NOW (PA, NA), STAI (State, Trait), PHQ-8 and WHOQOL-BREF (Physical, Psychological, Social, Environmental) were analysed using separate linear mixed models (LMM) where Group (Flavonoid, Control) and Time (Baseline, 2 weeks) were fixed factors. An unstructured covariance matrix was used to model the repeat effects of Time. Participant was implemented as a random factor. Significant interactions were explored using Bonferroni-corrected pairwise comparisons between and within groups.

## 3. Results

Demographic and dependent variable data can be observed in Table 1 and Table 3, respectively, where no significant baseline differences occurred between flavonoid and control groups.

Mothers in the high flavonoid condition reported significantly lower state anxiety at the end of the 2-week intervention compared to baseline (*p* < 0.01; Figure 1a). This effect was not observed in the control group (*p* = 0.58). No significant main effects of Condition or Time were observed.

Intervention-related effects were also observed in the Quality of Physical Health domain where participants in the high flavonoid condition experienced significantly higher perceived quality of physical health at the end of the 2-week intervention compared to baseline (*p* < 0.01). This pattern did not occur for control participants (*p* = 0.75; F(1,33.39) = 7.32, *p* = 0.01; Figure 1b).

There were four significant main effects of Time on Trait anxiety (F(1,33.15) = 20.07, *p* < 0.01), Quality of Physical Health (F(1,33.39) = 5.19, *p* = 0.03), Quality of Psychological Health (F(1,33.01) = 4.31, *p* = 0.046) and Quality of Social Relationships (F(1,35.51) = 5.11, *p* = 0.03). Trait anxiety was significantly lower for all participants post-intervention compared to baseline (mean change = −3.31). Whereas perceived Quality of Physical Health was rated significantly higher at the end of the intervention compared to baseline, regardless of group (mean change = 2.78), as was perceived Quality of Psychological Health (mean change = 2.82) and Quality of Social Relationships (mean change = 6.25). Across time, reductions in NA approached significance (mean change = −1.91; F(1,36.71) = 3.94, *p* = 0.055).

No significant effects were observed in PA, perceived Quality of Environment or PHQ-8 measures.

## 4. Discussion

The current exploratory study recruited a sample of new mothers who had a baby under 1 year old to measure current mood, anxiety, depression and perceived quality of life, before and 2 weeks after a daily dietary flavonoid intervention as part of a randomised, controlled trial. In mothers who incorporated additional high flavonoid foods into their diet, levels of state anxiety significantly reduced across time, suggesting that regular consumption of flavonoids may have alleviated feelings of anxiousness in transitory emotional states. Flavonoid-supplemented mothers also reported a significant improvement in perceived quality of physical health, a domain that incorporates measures of energy, fatigue, activities of daily living and dependence on medication.

Mothers in the control group who were instructed to continue with their normal diet did not show any change in anxiety scores. This suggests that regular intake of flavonoids from typical dietary sources may alleviate feelings of state-dependent anxiety in new mothers, and would be a relatively accessible, cost-effective and manageable change for new mothers and family units to adopt during a period in which they may be under psychological, physical and financial strain. Indeed, 67% of mothers scored above the clinical cut-off for anxiety in a postnatal population [40] (Table 1) and 64% of mothers displayed depression symptomology as stipulated by the PHQ-8 [46] (Table 1), indicating the sample were experiencing moderate-high levels of anxiety and/or depression at day 0 (baseline). Only 12 women had no symptoms of depression as stipulated by the PHQ-8, a contrasting difference to the 29 women reporting no psychological diagnoses. These proportions interestingly align with research that suggests rates of PND are higher than initially acknowledged, potentially as high as 60% [15]. In the current sample at least, depressive symptoms do appear to be common amongst a mostly diagnosis-free group. This supports the notion that mothers in the 0–12-month postnatal period may be psychologically ‘at risk’ and may experience symptoms of psychological disorders without full recognition, support, diagnosis or treatment. Further, this highlights potential for further research into the barriers of PND diagnosis and subsequent treatment.

It must be highlighted that the findings from this study are to be considered in the context of the COVID-19 pandemic. Research conducted during the first UK national lockdown in 2020 [52] found that state anxiety (assessed using the STAI state scale (STAI-S)) amongst new mothers was high with 61% of women in the 0–12 week postnatal period reporting a score of 40 or above, signifying clinically relevant anxiety [38]. Additionally, pandemic levels of anxiety were significantly higher than pre-pandemic levels of anxiety, assessed by comparison to a study using matched methods and recruiting a postnatal population of similar characteristics [53]. As the current study was conducted in the interim between the UK’s first and second national lockdowns, it is likely that anxiety remained high, and this may have contributed to higher-than-average state anxiety scores. Replication of this study in non-pandemic circumstances is therefore warranted.

It is important to note that the STAI used to measure anxiety did not include hallmark clinical (DSM-V) symptoms of generalised anxiety disorder (GAD; [19]) such as irritability, fatigue or restlessness. Results should not, therefore, be interpreted as a reduction in GAD symptoms specifically, but a reduction in problematic thoughts and feelings that may make an individual feel anxious in any one given state. Early motherhood can be viewed as a series of new situations or ‘states’ which may be prone to anxiety and so a measure of thoughts and feelings without physical symptoms is perhaps a more informative measure to use in a postpartum population. The STAI was used in the current study to capture participants’ state anxiety in this vulnerable period and has been recognised as a valid screening tool for depression and anxiety disorders in the postnatal period [40]. However, research has found that feelings of anxiety in new mothers are not necessarily generalised, as in non-childbearing populations with GAD, but are more maternal- and infant-focused, and include worries about maternal competency, mother-infant attachment, infant safety, practical infant care and psychosocial adjustment to motherhood [54]. It is important that future research acknowledges the distinction between GAD and postpartum anxiety and uses appropriate additional measures to capture anxiety in a postpartum population, such as the Postpartum Specific Anxiety Scale [54] alongside measures of state anxiety.

Presently, the flavonoid intervention was also associated with significantly higher perceived quality of life in the domain of Physical Health, whilst no change was observed for the control group. This measure encompasses seven facets including activities of daily living, dependence on medicinal substances and medical aids, energy and fatigue, mobility, pain and discomfort, sleep and rest and work capacity [48]. Depletion in these facets is common in new mothers, suggesting that the physical health domain may have a higher capacity for change. Currently, there are no WHOQOL normative data for non-patient, parent populations, so it is unclear if the change in physical quality of life observed in this study is clinically meaningful. Previous healthy young adult data (*n* = 240, mean age = 22.9 years (SD = 3), 47% male) established a standardised physical health mean score of 78.83 (SD = 13.09) [48], which is noticeably higher than standardised baseline scores of flavonoid (mean = 40.13, SD = 9.38) and control (mean = 44.89, SD = 8.48) participants in the current study (mean age = 29.21, SD = 5.67). Interestingly, physical health data in the current sample are closer to the standardised norms of caregivers of relatives with psychiatric illness (mean = 53.43 (SD = 14.2), *n* = 266, mean age = 37.3 years (SD = 11.3), 52.6% male) [55] and patients with type 2 diabetes (mean = 48.1 (SD = 20.4), *n* = 344, mean age = 40.54, SD = 15.21, 57.3% male) [56], suggesting that an all-female, new mother sample may have lower than average norms for quality of life, as indicated by the current results. Indeed, psychological, social and environmental domains also appear to be lower than published norm data [48,55,56]. Further studies recruiting a larger N should aim to establish normative data for this population. Forthcoming research should investigate what may be driving the improvement in physical quality of life, for example which physical factor in particular may be boosted by flavonoid consumption. The other measures of quality of life (social, psychological or environment) were not affected by the intervention. It is perhaps ambitious to expect a two-week dietary intervention to improve mothers’ perceptions of their social connections and overall living environment, but it is more plausible that changes in psychological quality of life could be detected. Given the observed benefits to anxiety, but absence of effects for depression, it appears that changes in psychological state are subtle, and the psychological quality of life measure is likely unsuitable for detecting these changes. It may be more sensitive to long terms changes such as several months or more.

The absence of differences between the flavonoid and control group, or indeed any changes over time for the measures of depression (PHQ-8) and positive affect implies that a 2-week dietary flavonoid intervention may not be sensitive enough to impact depressive symptoms and related affective components in a postnatal population. This is contrary to findings by Fisk et al. [10] who observed a significant reduction in depressive symptoms (using the Mood and Feelings Questionnaire) in adolescents following a 4-week WBB regimen. It is possible that two weeks is not long enough to detect changes in depressive symptoms. Furthermore, considering the current population, the PHQ-8 may not have been specific enough to assess symptoms of PND; PHQ-8 items are more focused on symptoms of MDD. Future investigations should consider using validated PND-specific scales (such as the Edinburgh Postnatal Depression Scale) to ensure that the full range of possible PND symptoms are measured. Of relevance here is that only three participants had a PND diagnosis in the current study. The absence of specific depression-related effects may be due to most participants who had depressive symptoms fitting in the ‘mild’ depression category at baseline (42%). Much lower prevalence was seen in the higher depression categories of ‘moderate’ (8%), ‘moderately severe’ (3%) and ‘severe’ (3%) where the potential for improvement in symptoms may be higher.

Interestingly, in addition to state anxiety reductions in the flavonoid group, significant reductions in trait anxiety and NA were observed over time for both groups, which may have been a by-product of taking part in a study which drew attention to individuals’ mood states. Awareness of mood can be therapeutic in itself [57] and, in retrospect, it would have been interesting to ask mothers how they labelled and interpreted their responses to baseline mood questionnaires and whether they explored any mood materials or self-help literature after taking part in the initial baseline session. Another possible methodological consideration is that the very act of consciously logging intake of a high flavonoid food could positively influence mood outcomes, by potentially highlighting a positive dietary behaviour. The control group were not asked to make any recordings of their diet during the 2-week period, and therefore this is an unavoidable confound within the design. The alternative was to ask the control group to report dietary intake during the 2-week period, however, this brings other limitations such as changes in dietary behaviour and potentially reduced compliance, which reduces the validity of a control group. Further, the use of dietary self-report measures may be methodologically limiting due to participants forgetting to log items, or by including or excluding particular items to appear ‘healthier’. Methods of dietary intake could be further improved by assessing urinary samples for polyphenolic conjugate concentrations; however, urinary procedures would be highly intensive for a sample of new mothers and may result in attrition. Methods that are more objective and less intensive should be utilized; for example, logging food weights or recording items via an app with the option to set reminders after meals. A further methodological improvement is the platform used to host the study questionnaires. Although user-friendly, Microsoft Forms had a cap on the number of questions allowed in one form, meaning the study was split across two links. This, unfortunately, resulted in some missing data across measures where participants did not submit Parts 1 or 2 correctly. Questionnaire software that allows a large number of questions should be used in the future.

When investigating mood across a postnatal population it is important to consider the biological changes that occur in pregnancy and in the immediate period after birth as potential sensitive windows for nutritional intervention. For example, it is known that a major brain change occurs in the immediate 5 days post-birth where a 43% increase in monoamine oxidase A (MAO-A) is observed in mood-modulating brain areas during postpartum blues (PPB) [58,59,60,61,62]. Severe PPB has been found to be a significant predictor of later clinical PND [63,64] suggesting postnatal days 0–5 as prodromal for the subsequent postnatal mood experience. Results from Dowlati et al. [65] show promising support for nutritional intervention in this vulnerable period. Specifically, mothers consumed a daily supplement containing 2 g tryptophan, 10 g tyrosine and 280 mL blueberry juice (Milne Fruit Products) with blueberry extract (1 g VitaBlue) across postnatal days 3–5. On day 5, mothers in the placebo group, who did not take a supplement, reported higher depressed mood on a Visual Analog Scale (VAS) following a sad mood induction procedure (MIP), whereas supplemented mothers showed no such effect. Such results imply that a concoction of monoamine precursor amino acids and dietary antioxidants may counteract mood disturbance in the PPB peak, which may reduce the likelihood of clinical PND. Forthcoming trials should consider an intervention in the postnatal period closer to birth where vulnerability to long-lasting mood change may be higher and should utilise a higher number of participants to improve power and sensitivity to detect mood effects. In addition, there is a need for more data on the natural course of depression and anxiety throughout pregnancy and the postpartum period, and exploration into the effects of nutritional intervention in the antenatal period is warranted. Thus, forthcoming research should aim to explore mood effects in flavonoid-supplemented populations pre- and postnatally to identify potential critical periods for positive mood change. In addition, research containing measures of oxidative stress, BDNF regulation, or other physiological markers would be informative alongside mood measurement to investigate the mechanistic potential of flavonoids as therapeutic agents in reducing or preventing mood disorders.

In conclusion, this research adds to the growing body of literature implementing flavonoids as beneficial dietary constituents for healthy mood. Specifically, this exploratory study demonstrated alleviated feelings of state anxiety and improved perceived quality of physical health in a community sample of 0–12-month postnatal mothers following a 2-week daily dietary flavonoid intervention.

## Figures and Tables

**Figure 1 nutrients-13-02383-f001:**
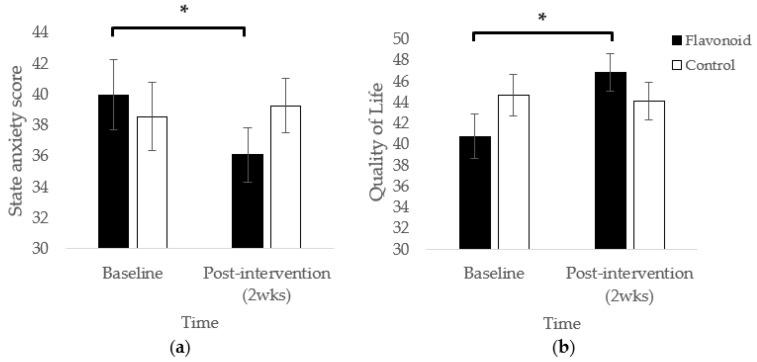
(**a**) Mean (SEM) state anxiety was significantly lower for mothers at 2 weeks compared to baseline (F(1,32) = 5.93, *p* = 0.02); (**b**) Mothers in the flavonoid group rated mean (SEM) quality of their physical health as significantly better at the end of the intervention compared to baseline (*p* < 0.01). * *p* = 0.01.

**Table 1 nutrients-13-02383-t001:** Demographic data for mothers and babies in flavonoid and control groups.

Measure	All Participants	Flavonoid Group (*n* = 19)		Control Group (*n* = 19)		Between Groups *p* Value ^a^
	Mean (SD)	Mean (SD)	Range	Mean (SD)	Range	
Age of mother (years)	29.21 (5.67)	29.58 (5.69)	20–41	28.84 (5.78)	20–41	0.70
Sleep of mother (h/night)	6.03 (1.15)	6.00 (1.25)	3–8	6.05 (1.08)	4–8	0.89
PHQ-8 symptomology ^1^	16:3:1:1	7:2:1:0	-	9:1:0:1	-	0.47
State anxiety symptomology ^2^	11:22	4:11	-	7:11	-	0.46
Age of baby (weeks)	21.5 (8.35)	23.11 (7.76)	5–35	19.89 (8.81)	5–43	0.24
Sex of baby (female:male)	20:18	10:9	-	10:9	-	1.00
Term of baby (very early:early:full:late) ^3^	2:12:18:6	0:6:9:4	-	2:6:9:2	-	0.45
Feeding method ^4^						
0–6 months	18:3:10:7	8:3:6:2	-	10:0:4:5	-	0.18
6–12 months ^5^	4:8:6:6	2:6:4:2	-	2:2:2:4	-	0.35
Complications (yes:no)						
Pregnancy	8:30	6:13	-	2:17	-	0.11
Birth	7:31	3:16	-	4:15	-	0.67
Breastfeeding	12:16	7:12	-	5:14	-	0.49
Habitual diet (items/day)						
Flavonoid ^6^	4.05 (2.52)	4.56 (2.44)	1–10	3.64 (2.59)	0–12	0.30
Junk food	1.85 (1.84)	1.61 (1.18)	0–5	2.09 (2.34)	0–11	0.43

^1^ Mild (score of 5): moderate (score of 10): moderately severe (score of 15): severe (score of 20+); ^2^ Below cut-off: above cut-off. A cut-off of 36 was used as it has been deemed predictive of disordered anxiety in a postnatal population [40]; ^3^ Very early term = born between 32–36 weeks; Early term = born between 37–38 weeks; Full term = born between 39–40 weeks; Late term = born at 41 weeks+; ^4^ Frequency of babies breastfeeding: formula feeding: combination feeding: prefer not to say; ^5^ Flavonoid *n* = 14, Control *n* = 10; ^6^ Mean flavonoid items per day; flavonoid *n* = 15, Control *n* =18. ^a^ Post-hoc achieved power for between group baseline comparisons using an effect size of *d* = 0.3 (small) was low (0.23); t (36) = 1.69.

**Table 2 nutrients-13-02383-t002:** Breakdown of the number of high flavonoid foods added to the diet per person across the 2-week intervention in the flavonoid diet group (*n* = 20). Units represent the average (SEM) number of occasions an item was consumed per person across 2 weeks.

High Flavonoid Food Item Added to the Diet	Mean (SD) Number of Occasions Item Was Added over 2-Week Period (Per Person)
Coffee (4–5 cups)	3.65 (3.86)
Berry fruits (~120 g)	3.45 (2.61)
Tea (4–5 cups)	3.40 (4.17)
Orange juice (250 mL)	1.75 (2.22)
Leafy green vegetables (~70 g)	1.10 (1.52)
Dark chocolate (2 large squares)	0.65 (1.09)
Red wine (250 mL)	0.55 (0.89)
Grapefruit juice (250 mL)	0.40 (0.88)

**Table 3 nutrients-13-02383-t003:** Mean (SD) raw outcome variable data for participants in the flavonoid and control groups at baseline and 2-weeks post-intervention.

	Baseline		Baseline *p* Value	2 Weeks		Change from Baseline		Interaction *p* Value
Measure	Flavonoid	Control		Flavonoid	Control	Flavonoid	Control	
PANAS-NOW	(*n* = 19)	(*n* = 19)		(*n* = 17)	(*n* = 18)			
PA	29.53 (6.88)	26.95 (8.42)	0.27	30.65 (4.82)	28.50 (6.42)	−0.78 (11.06)	−0.44 (6.90)	0.71
NA	19.58 (6.43)	20.11 (8.69)	0.89	17.39 (7.29)	18.39 (7.52)	−2.56 (4.24)	−2.61 (9.20)	0.59
STAI	(*n* = 15)	(*n* = 18)		(*n* = 18)	(*n* = 18)			
State anxiety	40.27 (9.66)	38.33 (10.59)	0.84	36.56 (5.86)	39.06 (9.39)	−4.21 (5.49)	0.82 (5.43)	0.02 *
Trait anxiety	44.53 (7.61)	41.94 (7.57)	0.29	40.50 (7.65)	38.72 (7.97)	−3.42 (2.59)	−3.00 (5.32)	0.72
PHQ-8	(*n* = 15)	(*n* = 18)		(*n* = 17)	(*n* = 18)			
	6.47 (4.64)	5.56 (5.19)	0.60	5.65 (3.35)	5.06 (4.33)	−0.62 (3.50)	−0.12 (3.64)	0.80
WHOQOL-BREF	(*n* = 15)	(*n* = 18)		(*n* = 17)	(*n* = 18)			
Physical	40.13 (9.38)	44.89 (8.48)	0.14	47.06 (8.79)	43.94 (6.93)	5.62 (8.31)	−0.29 (6.01)	0.01 *
Psychological	49.73 (11.20)	55.00 (9.24)	0.15	54.53 (11.47)	56.44 (9.41)	4.23 (8.24)	1.18 (7.58)	0.24
Social	55.27(18.77)	55.72 (16.98)	0.94	61.94 (13.74)	61.28 (17.75)	7.15 (13.96)	5.53 (18.04)	0.81
Environment	66.73 (11.57)	66.44 (14.43)	0.95	66.35 (9.47)	66.11 (9.94)	1.00 (8.84)	−1.06 (14.65)	0.92

PANAS-NOW: Positive and Negative Affect Schedule; PA: positive affect; NA: negative affect; STAI: State-Trait Anxiety Inventory; PHQ-8: Patient Health Questionnaire 8; WHOQOL-BREF: World Health Organization Quality of Life (abbreviated version). * represents significance (*p* < 0.05) following Bonferonni adjustments.

## Data Availability

Data is contained within the article and available upon request to the corresponding author.

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
