# Peer review of "Mental Health in New Mothers: A Randomised Controlled Study into the Effects of Dietary Flavonoids on Mood and Perceived Quality of Life"

_nutrients, 2021, doi:10.3390/nu13072383_

Round 1

Reviewer 1 Report

I agree with the comments provided by the other reviewer and recommend that the manuscript be accepted for publication.

Reviewer 2 Report

Yes edits area adequate. Thank you

Reviewer 3 Report

The authors accepted the suggestions and made the appropriate modifications, adding further information and enriching the final version. I consider this manuscript very complete, detailed and well written. Therefore, I believe the manuscript ready for publication.

This manuscript is a resubmission of an earlier submission. The following is a list of the peer review reports and author responses from that submission.

Round 1

Reviewer 1 Report

Review of the article

“Mental health in new mothers: a randomized controlled study into the effects of dietary flavonoids on mood and perceived quality of life”

            Thank you very much for the opportunity to review this interesting trial linking maternal nutrition to maternal well-being after birth. While I consider the research question relevant, I have some minor and major concerns with the conducted study and analyses.

Major concerns:

  1. My biggest concern is related to the sample size, power analysis, and results. The authors did not justify their assumed effect size of 0.22. They also do not state what type of effect size was used (I assume it is f). An effect size of f=.22 to me seems rather large. However, the authors have nine dependent variables that were separately analyzed. As the authors applied a Bonferroni correction, my replicated power analysis with α = .006 (.05/9) resulted in 72 participants (all other assumptions being equal). In addition, the sample size is too small to identify differences between the groups at baseline. For instance, although non-significant, the group difference in flavonoid diet at baseline would be of d≈.36, resulting in an achieved power of just 20%. Finally, with regard to the results presented in Table 3 it is unclear to me whether a Bonferroni correction was actually applied as stated, as none of the significant p values were below the threshold of α = .006.
  2. Related to the first point, it is unclear from the introduction, why the authors assessed anxiety and health related quality of life. The introduction is about post partum depression. Especially with an unclear Bonferroni correction and without a registered study protocol, these results seem a bit fishy, although I highly respect the authors for probably stating all their dependent variables. In addition, although the authors separately discuss the presence or absence of effects on their dependent variables, they did not discuss the pattern. It is a difference of finding zero effects on depression and finding zero effects on depression given an effect on state anxiety. Given the presented pattern, the reported significant effects might more likely be due to chance than resembling real effects.
  3. The authors themselves acknowledge in the discussion that the effects might not be related to food intake but to the diary method. I agree with the authors that having the control group to fill out a diary might change their dietary behavior. However, I believe that still would have been the much better control group. Given the current design it is impossible to attribute any possible effects on dietary differences – especially as we do not know the actual dietary behavior in the control group. If the authors do not have the possibility to recruit another control group, they at the very least should refrain from attributing effects to nutrition (already the title states an effect of flavonoids) and abandon any language of cause or effect. However, I would be more convinced if the authors could claim a dose effect in the experimental group (e.i., stronger effects for mothers being more compliant to the intervention and consuming more flavonoids).

Minor concerns:

  1. I would suggest to move parts from the discussion in the introduction (related to the biological reasoning why flavonoids might effect mood).
  2. Please add numbers and references in the sample description that the rates of women with psychological disorders reflect the expected prevalence rates.
  3. As the study was conducted during the Covid-19 pandemic, I would expect the authors to discuss possible Covid associations. In particular, anxiety could be influenced by the pandemic. Keeping a diary could especially be helpful to restore control and therefore, reduce symptoms of anxiety.
  4. Given that the small sample size might have resulted in dietary differences at baseline, I would be much more convinced if the effects remain after additionally controlling for baseline habitual flavonoid consumption.
  5. Please, report the internal consistencies of the questionnaire measures from the present study.
  6. Please, add effect sizes for the reported effects.
  7. There seems to be a contradiction when the authors state “64% of mothers displayed depression symptomatology” (p.10) but claim that the “absence of specific depression-related effects may be due to the relatively low or mild scores of depressive symptoms”. (p.11) Please, clarify these statements.

Reviewer 2 Report

This paper investigates flavanols as a diet intervention to improve mood in n=41 postpartum women. The methods are largely adequate, though see below. Results are modest. The study of alternative forms of intervention for perinatal mood and anxiety disorders is an important topic; this study with a small n modestly adds to this work. It should be framed as an exploratory study.

  1. There is not a consensus that the postpartum period is a time of increased vulnerability to depression. See Olfson, M. et al. on this.
  2. Need more data on natural course of depression throughout first 12 months of pregnancy given question want to ask; See Samantha Meltzer Brody on # of women who ‘convert’ to depression from distress in pregnancy, # PPD w/o prior symptoms.
  3. Baby blues defined as physiological event is inaccurate; what mood state is not also a physiological event? Seen Ken Kendler on Toward an Integrated Philosophical Approach in Psychiatry.
  4. Baby’s have a sex, not a gender
  5. That the control group was not monitoring their food consumption is unfortunate? They could have spontaneously changed
  6. Which researchers WERE blind to status? This matters for interaction with particpants
  7. Good use of cut off scores
  8. Table 1, why not list range of scores even though have SD?
  9. Analyses do not seem set up to compare the difference in change? Why are baseline values in table 2 different from table 1? Do not need baseline mood in both places
  10. Physical activity results may be significant but are not clinically meaningful, are they?
  11. Were corrections for multiple comparisons done? Mom age at time of intervention still could interact with intervention to affect outcomes.

Reviewer 3 Report

The topic on the effect of flavonoids on mood and decreased risk of developing depression in recent years it has attracted great interest from researchers. I have read the manuscript with interest. The topic falls in the scope of this journal. I found this manuscript very complete, detailed and well written. The design is reasonable and the authors have given good number of citations about the subject.

I just give a suggestion to the authors about the “Keywords”. It is better to avoid repetition of words already present in the title. By using words not present in the title, you increase the possibility of finding it in articles searches. For example, I suggest to replace “flavonoids” with “phytochemicals” or “polyphenols”.

Reviewer 4 Report

Thank you for the opportunity to review the manuscript titled Mental health in new mothers: a randomised controlled study into the effects of dietary flavonoids on mood and perceived quality of life. This randomised controlled trial examined the effects of a 2-week dietary flavonoid intervention on 41 new mothers’ mood, anxiety, depressive symptoms and quality of life. The study was very well designed, accounting for previous dietary intake and using creative strategies for blinding. The data analysis was appropriate and the results are clearly presented in the text, tables and figures. The manuscript is professionally presented, written clearly and containing all relevant information. The introduction clearly demonstrates a rationale for the research and the discussion draws on relevant literature, discusses practical and mechanistic issues, and presents several suggestions for pertinent future research. I congratulate the authors on an excellent study and manuscript.

I have one suggestion for improvement of the manuscript. A small section in the discussion addressing the limitations of the study such as small sample size, missing data and use of self-report measures is warranted.